# A Non-Fungible Token Solution for the Track and Trace of Pharmaceutical Supply Chain

Ferdinando Chiacchio * , Diego D'Urso, Ludovica Maria Oliveri , Alessia Spitaleri, Concetto Spampinato and Daniela Giordano

Department of Electrical, Electronic and Computer Engineering, University of Catania, 95125 Catania, Italy; ddurso@diim.unict.it (D.D.); ludovica.oliveri@phd.unict.it (L.M.O.); alessia.spitaleri@unict.it (A.S.); concetto.spampinato@dieei.unict.it (C.S.); daniela.giordano@dieei.unict.it (D.G.)
* Correspondence: chiacchio@dmi.unict.it

**Abstract:** Industry 4.0 is leading society into a new era characterized by smart communications between consumers and enterprises. While entertainment and fashion brands aim to consolidate their identities, increasing consumers' participation in new, engaging, and immersive experiences, other industry sectors such as food and drugs are called to adhere to stricter regulations to increase the quality assurance of their processes. The pharmaceutical industry is inherently one of the most regulated sectors because the safety, integrity, and conservation along the distribution network are the main pillars for guaranteeing the efficacy of drugs for the general public. Favoured by Industry 4.0 incentives, pharmaceutical serialization has become a must in the last few years and is now in place worldwide. In this paper, a decentralized solution based on non-fungible tokens (NFTs), which can improve the track and trace capability of the standard serialization process, is presented. Non-fungible tokens are minted in the blockchain and inherit all the advantages provided by this technology. As blockchain technology is becoming more and more popular, adoption of track and trace will increase tremendously. Focusing on the pharmaceutical industry's use of track and trace, this paper presents the concepts and architectural elements necessary to support the non-fungible token solution, culminating in the presentation of a use case with a prototypical application.

**Keywords:** track and trace; serialization; blockchain; smart manufacturing; supply chain; Web3

## 1. Introduction

Industry 4.0 (I4.0) is playing a fundamental role in enhancing industry competitiveness with new technologies and digital applications that help improve the management, effectiveness, and sustainability of the production and distribution processes at different levels of the enterprise organization. To this end [1], I4.0 promotes a holistic methodology [2], by applying emerging technologies to revolutionize the current production and transit from machine-dominant manufacturing to digital manufacturing.

The main characteristic of this revolution is the depth at which the latest digital technologies, such as cyber-physical systems, Internet of things, big data, blockchain, and cloud manufacturing, are being integrated at several levels of an enterprise and, above all, how such a paradigm is gradually becoming extended to the supply chain, allowing greater communication, sharing, and organization of the data and activities among the enterprises.

As discussed in [3], I4.0 applications could help reduce production costs by 10–30%, logistic costs by 10–30%, and quality management costs by 10–20%. In [4], other advantages are identified. They include:

- A shorter time to market for the new products (using, for example, 3D printing, ERP, virtual manufacturing, and MES);
- An improved customer responsiveness;

- Enabling a custom mass production without significantly increasing overall production costs;
- More flexible and friendlier working environment (due to robotics, M2M, etc.);
- More efficient use of natural resources and energy.

Moreover [5], as the applications and use cases of I4.0 increase, what has emerged is that new paradigms and management styles for conducting the business of the enterprises have been proposed, especially in the product development [6], manufacturing activities, conception of new services [7], improvement of ergonomics [8], and management of the supply chain. Among the above, the track and trace process of the pharmaceutical products is analysed in this paper.

Track and trace of pharmaceutical products is very important: it has been estimated that counterfeit drugs cause over 100,000 deaths annually and over $200 billion of loss in the pharmaceutical industry [9]. To fight this phenomenon and prevent falsified medicinal products from entering the legal distribution chain, the 2011/62/EU Directive [10] introduces new regulations for pharmaceutical industries and distributors to enable a process of serialization that involves all the actors of the supply chain in the operations of production, packaging, tracing, and distribution. At the current state of the art, the centralized architecture on top of this regulation is not free from weaknesses and it fails at fulfilling all the requirements for the implementation of robust anticounterfeiting measures. The latter, on the other hand, can be achieved by integrating the current serialization technologies of I4.0 with other paradigms, such as blockchain and non-fungible tokens (NFTs), that favour the interoperability, communication, and trust among the actors of the distribution network up to the final consumers.

This paper is organized as follows: in Section 2, a literature review framing the state of the art of Industry 4.0 applications is discussed; a breakdown of the main technologies, advantages, and complexities in several industrial fields is thus presented. Section 3 introduces the main concepts of the track and trace and serialization process for the pharmaceutical industry, highlighting what are the issues that can be overcome by exploiting the integration of cutting-edge concepts such as blockchain and non-fungible tokens (NFTs) together with Industry 4.0 applications. Section 4 describes the main concepts behind the blockchain technology, including Web3 and NFT. The use case of a prototype developed exploiting the VeChain Thor blockchain technology is presented and the main enhancements achieved by adopting the NFT serialization process are summarized. Finally, conclusions are discussed in Section 5.

## 2. Literature Review

According to [1], "Industry 4.0 (I4.0) defines a methodology to generate a transformation from machine dominant manufacturing to digital manufacturing". In their literature review, six design principles of I4.0 were defined: interoperability, virtualization, local, real-time talent, service orientation, and modularity. The driving technology of I4.0 are smart sensors, artificial intelligence (AI), robotics, drones, nanotechnology, cyber-physical systems (CPS), additive manufacturing (3D printing), big data analytics, cloud computing, Internet of things (IoT), enterprise resource planning (ERP), machine-to-machine communication (M2M), and manufacturing execution system (MES) [11,12]. These technologies are integrated with each other in the smart industry.

As shown in Table 1, no matter the business sector, I4.0 provides great benefit to the whole process of setup, production, and lifecycle of an enterprise, including the management of the supply chain, demand prediction, and systems maintenance. In the following, a short summary of the most representative technologies of I4.0 is presented, including the industrial and business sectors in which they are intervening.

**Table 1.** Breakdown of the main technologies of I4.0 adopted in different industry sectors.

| Technology | Strength | Disadvantages | Industry Sector | Paper |
|---|---|---|---|---|
| Internet of things | Process control, increase yield, maximize productivity, enhance information flow | Security issues, securitypolicy | Pharmaceutical industry, food industry, construction industry | [13] [14] [15–18] |
| Cyber-physical system | Easier access to information, preventive maintenance, decision making, optimization routines | Security issues | Logistics | [19,20] |
| Machine-to-machine communication | Easy monitoring of resources and production lines, improve resources reusing, reduces operational costs, automate the decision process, favour a human free manufacturing environment | | Smart agriculture, smart grid, smart environment control, | [21] |
| Cloud system | Reduces costs, eliminates infrastructure complexity, extends work area, protects data, provides holistic access to information, increase speed and quality of production | Data integrity and availability | Medical service industry | [22] |
| Cloud computing | Allows real-time collaboration from different locations, enhance decision-making, and ensure project deliverability | | Construction industry | [15,23] |
| Big data | Provides business value through better strategic and operational decisions | Privacy law, perception of risk and unreliability of open data movement | SME Finance Logistics | [24,25] [26] [27] |
| Augmented reality | Aids the design phase of products and production systems, reduce time to market and cost | Social impact | Medicine Interior design Fashion retails Museums Shipyard manufacturing system | [28–30] [31–33] [34–36] [37–39] [40–42] |
| Enterprise resource planning | Improves process control, early indication of fails, communication transparency, optimize material and human resource utilization | Interoperability | Food industry Stone industry | [43,44] [45] |
| Virtual manufacturing | Shorter lead time, reduced cost, more efficient and improved quality with clean and green process | | Metal forming Manufacturing industry | [46] [47–50] |
| 3D printing | Design and print parts | | Archaeology Medical service industry Mechanical industry | [51] [52,53] [3,54] |
| Intelligent robotics | Reduces human force, inspection of dangerous process | Industrial accidents, human unemployment | Manufacturing industry Medical service industry | [55] [56,57] |
| Blockchain | Security, traceability, immutability, accessibility of data provenance | No interoperability, data privacy policy, immutability | Pharmaceutical supply chain | [58] |

### 2.1. Internet of Things

The Internet of things (IoT) is the internetworking of physical devices, vehicles, buildings, and other items embedded with electronics, software, sensors, actuators, and network connectivity that enable these objects to collect and exchange data. It consists of four major layers: perception layer, network layer, support layer, and application layer [20,59]. One weakness of the IoT could be security issues and cogent laws regarding privacy. In [14], the potential benefits and opportunities for using IoT architecture in planning, management, and control of the food supply chain (FSC) operations are presented. As discussed in [13], the IoT possibly supports pharmaceutical and gadget organizations in improving quality yield, reducing costs, and changing how medicine is delivered to the prescribers. According to [15], for the construction industry, the IoT can maximize productivity [16], enhance information flow during a project lifecycle [17], optimize energy efficiency [18], and improve security, safety, planning, managing, and monitoring of resources if it is coupled with building information modelling (BIM), a central repository for collating digital information about a project.

### 2.2. Cyber-Physical Systems

Cyber-physical systems (CPSs) refer to the integration of computing and physical processes. These systems can be divided in two elements: (i) a network of objects and systems communicating with each other through Internet with a designated address, and (ii) a computer simulation of real objects and their behaviours in a virtual environment.

Cyber-physical systems provide functionalities such as easier access to information, preventive maintenance, predefined decision making, and optimization routines; on the other hand, they have some security problems [15–17].

### 2.3. Machine-to-Machine Communication

Machine-to-machine (M2M) communication refers to direct communication between devices that can be wired or wireless. According to [60], it can include industrial instrumentation, enabling a sensor or meter to communicate the data it records to application software programmed to use it. The main advantages, as summarized in [1], reveal that M2M communication allows easy monitoring of resources and production lines, helps to better use the resources and reduce operational costs, increases the ability to automate the decision process, and allows for the creation of a human-free manufacturing environment.

Moreover, in [21], the possibility of using smart sensors to enable smart environments (e.g., monitoring of weather pollution, monitoring of radiation levels, monitoring of electromagnetic levels, smart lighting systems, noise mapping of the city, and waste management), smart agriculture (e.g., more efficient agricultural production and environmental protection), and smart grids (e.g., power management system) is discussed.

### 2.4. Cloud System and Cloud Computing

According to [61], cloud technology is the simplest online storage service that provides web-based applications that do not require any installation. It facilitates operation by ensuring that customers and employees reach the same data at the same time. Cloud systems reduce costs, eliminate infrastructure complexity, extend work area, protect data, provide access to information at any time, and increase the speed and quality of production [62]. The system proposed in [63] can generate knowledge from data collected in order to improve a manufacturing environment.

On the other hand, the system of storing all applications, programs, and data in a virtual server is called cloud computing. It can provide access to large datasets and clusters of remote processors to filter, model, optimize, and share data across systems to improve performance over time. The integration of BIM into cloud computing allows project stakeholders to collaborate in real time from different locations to accelerate decision making and ensure timely project delivery [15,23].

### 2.5. Big Data

Big data is essential for processes and has a remarkable impact on systems [20–25]. According to [64,65], the goals of big data are: To analyse and summarize data for pattern recognition and classification (descriptive), make accurate predictions of the monitored asset (predictive), or provide business value through better strategic and operational decisions (prescriptive). Big data is collected from a lot of sources at an increasing velocity, volume, and variety. Operators need an optimal processing power, analytics capabilities, and information management skills to extract value from big data [66]. According to [1], the implementation of big data analysis is not easy, mainly for privacy law and for perception of risk and unreliability of open data movement.

### 2.6. Smart Factory

Smart manufacturing, or dark factory, is a category of manufacturing aiming to optimize concept generation, production, and product transactions from traditional approaches to digitized and autonomous systems [67]. In a smart factory, there is little intervention of human power. Pillars of a dark factory are virtual reality, augmented reality, simulations, and virtual prototyping [68].

### 2.7. Augmented Reality

According to [69], "augmented reality is a technology that overlays digital information on objects or places in the real world for the purpose of enhancing the user experience". The main benefit of this technology is that it aids the design phase of products and production systems. In fact, creating the product and manufacturing layout in a virtual reality prevents errors [70]. As discussed in [71], augmented reality can be using in marketing and brand recognition, adding value. Moreover, it is also being implemented in medicine, museums, fashion, interior design, and numerous other areas.

### 2.8. Enterprise Resource Planning

Enterprise resource planning (ERP) is a generic name given to information systems designed to integrate and efficiently use all the resources of an enterprise. ERP software is a system that assists an enterprise in bringing together processes and data that are executed in all processes from sales to accounting, from production to human resources, or from stock management to purchasing. In [45], some advantages of ERP for the stone industry are presented, and [44] determined and classified the benefits of ERP system implementation in the dry food packaging industry. Among them, it is necessary to mention:

- Mobile applications may use ERP data to send messages to the manager and to the machines running in manufacturing;
- Real-time data can be aggregated and optimized for any batch size, analysed, and could allow early indications of fail or process drift, helping the preventive maintenance [72];
- ERP systems could allow for the access of information to suppliers, customers, and other partners to assure the efficiency of online operations and sales and purchasing transparency;
- Optimum material and human resource utilization could be possible;
- Customers may be able to track the status of their orders online.

### 2.9. Virtual Manufacturing

Virtual manufacturing (VM) is the use of computers to model, simulate, and optimize the critical operations and entities in a factory plant [47,73,74]. The main technologies used in VM include computer-aided design (CAD), 3D modelling and simulation software, product lifecycle management (PLM), virtual reality, high-speed networking, and rapid prototyping. Advantages of 3D printing are the possibility to print spare parts, industrial equipment, and articles used in daily life. In medical and surgical applications, 3D printers are used to create artificial organs [52,53]. According to [75], VM can reduce manufactur-

ing risks, improve manufacturing design and operation, support manufacturing system changes, enhance product service and repair, increase manufacturing understanding, and provide a vehicle for manufacturing training and research.

### 2.10. Intelligent Robotics

Robots can carry big things, work in dangerous conditions, and perform repetitive operations. Nowadays, robots have an important effect on society and [55] the biggest one is the risk of human unemployment. One proposed solution is that workers' profiles will change and those with capabilities aligned with the changes and progress will have new opportunities. On the other hand, numerous are the advantages of robotics, as they favour collaborative solutions to improve the efficiency of the system with little human intervention.

### 2.11. Blockchain

According to [58], "blockchain is a decentralized, immutable shared ledger that can be applied to a variety of business settings involving transaction processes". Advantages of blockchain are speed, security, traceability, immutability, transparency, and accessibility of data provenance. Blocks are connected by cryptography; changing one block would cause a rework from the earliest to the latest transaction in blocks, as illustrated by [76]. This is an advantage but also a limit, because in the event of mistakes, it can be quite difficult to correct them. Other limits are generated by the interoperability among different blockchain networks and the issues of data privacy (people might refuse to have their data stored permanently on the blockchain). On the other hand, blockchain technology is an appropriate solution for several critical application domains such as healthcare, transportation, agriculture, education, forecasting, etc.

In the pharmaceutical industry supply chain, due to cryptography, traceability can improve and mitigate the issues given by counterfeit drugs [77]. This last matter is the objective of this paper and is discussed in detail in the next sections. In [58], a decentralized solution for the tracking and tracing of the pharmaceutical drugs along the supply chain is presented. This solution is based on the Ethereum blockchain, which is the most famous public permissionless blockchain. Due to this great feature, the Ethereum blockchain is used and abused (as it is at the basis of tons of applications), and this decreases its performance while increasing the transactions costs. For this reason, the Ethereum blockchain is not the most appropriate for a supply chain application and stakeholders should look for a different blockchain platform. As it will be shown in the next sections, the VeChain Thor blockchain, a public permissioned blockchain, will be examined.

## 3. Track and Trace and Serialization Process of a Pharmaceutical Factory

Track and trace is one of the most important and valuable properties of advanced supply chain operations. It refers to the capability of monitoring a product along its lifetime, particularly until it is delivered to the final user. In particular, tracking refers to the ability to follow the path forwards from the starting point to wherever the object currently is, whereas the tracing of an object refers to the ability to follow the path backwards from its current point to where it began. It is often difficult to achieve without a technological infrastructure; therefore, it is generally dedicated to certain types of goods or services for which it is possible to keep track of all the locations and users that have handled it. In fact, both processes rely on a consistent system of identification that allows for a guarantee of the uniqueness of the identification of the object. For the pharmaceutical market, both these processes are very important and, nowadays, the implementation of a robust process of track and trace is more affordable due to the technological progress of automation and, in particular, the Industry 4.0 revolution.

In regards to the pharmaceutical industry, one of the main advantages of an effective traceability is that it can enable robust pharmacovigilance measures by favouring a prompt, accurate identification and mitigation of the specific products, manufacturers, and batches

of a biologic that can be the cause of potential adverse drugs reactions [78]. To this end, each actor of the supply chain distribution has to maintain lots of information for several years, in particular, the one related to their specific responsibility which pertains to their handling procedures. For the pharmaceutical manufacturers, this duty refers to the activity of generating a digital version of the batch record (electronic batch record), which, according to the good practice quality guidelines and regulations (GxP), implement an efficient data retention and archival policy to keep all the production information of a lot of products secure and available. The technological solutions that allow the automation of the required procedures to produce the electronic batch record [79] have already been available for many years, due to the diffusion of integrated software systems that are able to communicate and control the production layers of the factory. Among these, enterprise resource planning (ERP) and manufacturing execution systems (MES) are the most evolved [80]. An ERP system aids in processes such as sales functions, batch production, quality assurance, inventory control, and accounting, increasing the flexibility in the production process and improving the supply chain and distribution collaboration, whereas the MES acts as a central system with effective interoperations with other manufacturing systems and departments including the operations, quality, maintenance, and inventory control.

As previously stated, together with traceability, tracking can become an essential feature of the pharmaceutical market as it offers a foundation for an improved, trust-based relationship framework among producers, consumers, and all the actors of the distribution network. In these last few years, this process has gained the attention of researchers and practitioners because, due to digitalization, Internet of things, smart devices, and sensors, its feasibility with real-time up-to-date information has become affordable.

Figure 1 shows the implementation of an effective tracking process, involving not only the pharmaceutical manufacturer, but also all the other stakeholders of the supply chain, before dispatching to the final customer. The prerequisite for the implementation of the 2011/62/EU Directive, known as serialization, is that all the prescription drugs must have a unique identifier that allows for quick identification; in this way, the safe removal of counterfeit drugs and falsified medicines can be more effective [81]. From 2019, serialization has become a crucial feature because pharmaceutical companies not complying with the requirements for pharmaceutical serialization cannot sell medicinal products on the European market.

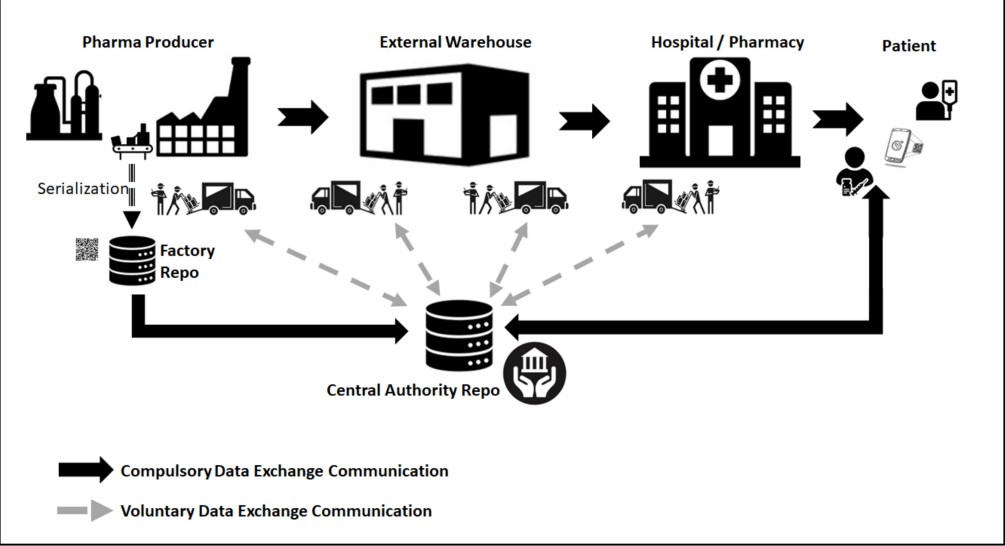

**Figure 1.** Track and trace process with a central regulatory.

The process requires that producers have to communicate all the serial numbers of the product units that have been produced and are ready to enter in the distribution network to a central authority. Safety, security, confidentiality, resilience, and reliability related

to the quality of the product and its transactions throughout its lifetime and distribution chain must be guaranteed by the central authority. In Europe, the central authority for the traceability of drugs is the European Medicines Verification Organisation (EMVO).

### 3.1. Serialization Process

As previously stated, the prerequisite for the implementation of the serialization process is the adoption of a unique code that allows the quick identification of a package and the product in which it is carried. Three main packaging layers can be identified in the pharmaceutical industry, as shown in Figure 2.

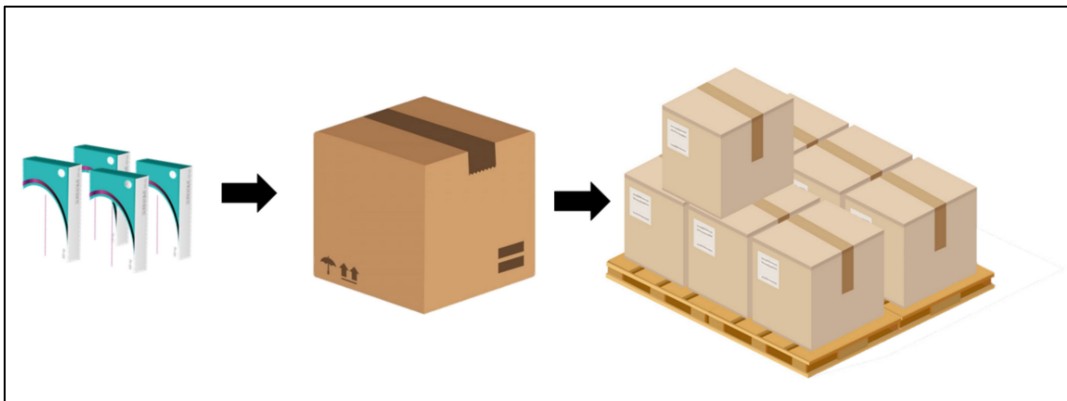

**Figure 2.** Serialization and packaging hierarchy (unit box, case box, and pallet).

**Primary Packaging** is the packaging in contact with the drug. The most notable examples of primary packaging is the aluminium blister suitable for hard packing of medical and pharmaceutical products such as pills, capsules, and tablets. Other examples are vials and flacons. The serialization of the primary packaging is not a requirement in any market except for those in which the patient can directly purchase the primary packaging.

**Secondary Packaging** contains the medicine or medicines inside the primary packaging. A carton box, known also a unit box, is the best example of this packaging. The serialization of this packaging is a must for regulations compliance; for the EU-FMD and other regulations, an additional tamper-evident mechanism is also required to avoid counterfeiting and guarantee that the medicine has not been opened before handed to the patient.

**Tertiary Packaging** is used to carry out business-to-business operations in the supply chain. Tertiary packaging includes pallets, bundles, and cases. This level of packaging is the most important point to ensure traceability with serialization.

In serialization, the packaging has to follow a hierarchical aggregation because the marking of each box with a unique identifier allows for dating back from the outer to the inner packaging level and vice versa, assuming that the packaging hierarchy is known and tracked in a computerized database. In fact, unit loads are not assembled randomly but according to the information carried by a work order [77]. To be compliant with the serialization specifications described in the 2011/62/EU Directive, each box of each level must be marked with a unique identifier. The codification of the identifier must respect several rules that depend on the packaging level (e.g., unit box, case box, or unit load) of the product packaged and, more importantly, on the encoding specification of the country in which the product will be traded.

The structures of the unique code definitions of each country are similar to each other and, in fact, following the GS1 standard, they are printed in the form of a 2D QR code. Because each country adopts slightly different QR code formats, in order to ship and commercialize a drug product, the pharmaceutical companies have to prepare the packaging of the products accordingly. Using the GS1 standard, the basic pieces of information required to create the QR codes are the following:

- *GTIN*: The global trade item number (GTIN) can be used to identify a product at any packaging level, e.g., consumer unit, inner pack, case, or pallet. This information provides a common language to uniquely identify the item worldwide for all relevant entities and trading partners;
- *Serial number*: A numeric or alphanumeric character sequence consisting of up to 20 digits, which must be unique for each GTIN;
- *Lot information*: A company-specific production that allows identification of the information of the lot produced;
- *Expiration date*: The expiration date of the product.

The GTIN can be obtained from the company that needs to commercialize a product by registering its global location number (GLN) with the local GS1 organization of the country. The global location number is a 13-digit number used to uniquely identify any legal entity, functional entity, or physical location. This unique identifier is comprised of a GS1 company prefix, location reference, and check digit. The GLN is used to uniquely identify any location or legal entity anywhere in the world. Figure 3 shows two examples of standard QR codes for Europe and the U.S.

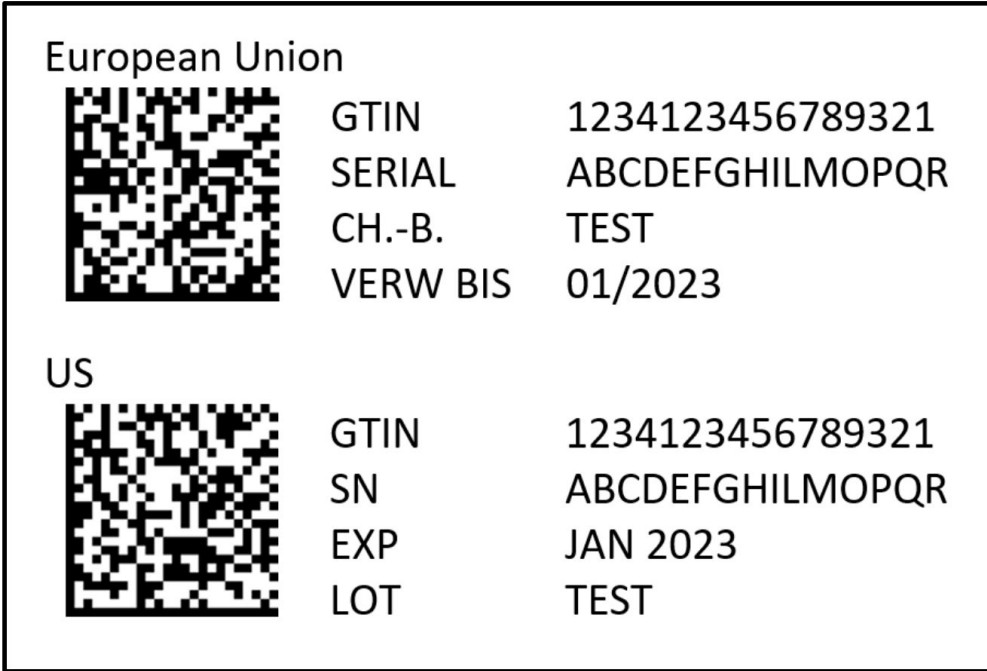

**Figure 3.** GS1 QR codes for the serialization standard in Europe and the U.S.

### 3.2. Serialization Manufacturing Technology

Among different countries, there can be differences in the serialization rules; therefore, in order to implement the correct pharmaceutical serialization, the producer has to adopt the correct serialization format when printing and applying the QR code on the packages where the product has to be commercialized. While the GLN is uniquely assigned to the producer industry and the GTIN is assigned to a specific product, the other information needed to create a QR code are dynamic and vary according to the batch of production and the country of destination. Therefore, in order to complete the serialization process, the manufacturer has to rely on a complex process of production that, in the context of the pharmaceutical producers, brings us to a production infrastructure made up of five technological levels, as shown in Figure 4.

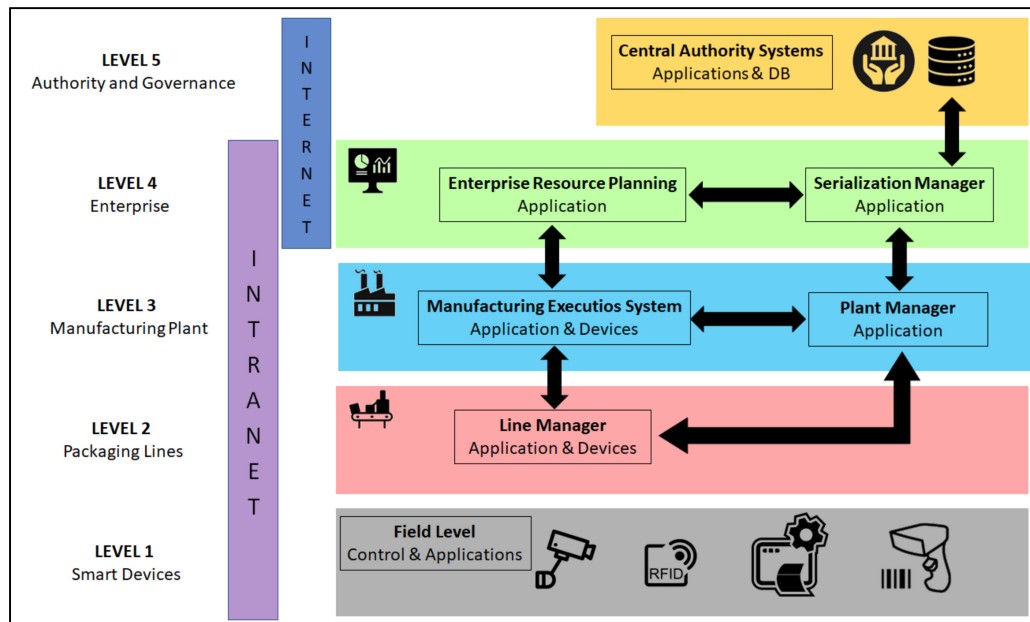

**Figure 4.** Architecture of the information system for the serialization process.

The industry equipment cooperates together at the various levels of the production process, allowing the automation of the packaging lines, which are generated as output pallets that aggregate a list of case boxes and unit boxes. The organization of the pallet is stored in the ERP system, which can be easily queried to identify the serial numbers characterizing each item. At the enterprise level (Level 4), the ERP and the serialization system communicate to exchange information about the next lot of products to schedule. This information is crucial to informing the equipment of Level 1 and Level 2 about Level 1, which communicate with each other and are controlled by the MES and/or the ERP. The cyber-physical systems that constitute the smart production and packaging lines are thus monitored by a real-time vision system that interacts with the other software (such as the ERP and MES), which are in charge of scheduling the lots of products to process according to a production plan.

The packaging phase is one of the most critical because if the QR code applied at the various package levels is not printed in the correct way, the item has to be discarded. Therefore, Level 4 instructs the printer of Level 2 and Level 1 to correctly set up the format of the QR code and the serialization numbers to apply to the packages. As previously stated, traceability requires a unique code for each package; this information has to be readable along the entire supply chain lifecycle. During packaging, it is necessary to check the quality of the QR codes printed on the packaging in order to guarantee that the items are correctly labelled. Therefore, the serialization solution requires the integration of a real-time vision system able to monitor the process and to communicate with the other software of the serialization infrastructure (such as the ERP and MES) and with the packaging lines. This way, if the label of a serial number that characterizes a specific box unit is not correctly printed, it is automatically discarded.

Another important function of the vision system is to control the aggregation requirement to guarantee that the hierarchical packaging performed by the packaging lines is correctly handled. In other words, the vision system keeps track of the boxes that have been placed into each case and the cases that have been placed into each unit load. As soon the last level of packaging has ended (generally corresponding with the lot unit load), the serialization software sends the hierarchy matrix list to the ERP, which closes the work order and commences the beginning of the next phases (storage, batch release, and delivery and shipping). During the batch release, the serialization software sends the official hierarchy

matrix list to the central regulatory (EMVO); it has to be stored permanently in the database of the serialization software.

The most commonly used method for serialization in the pharmaceutical industry is a combination of thermal inkjet with labelling technology, which guarantee high performance in terms of speed resolution of the QR code to print and stick. In fact, labelling can be used on several surfaces, such as serum bottles, nylon bags, etc., which cannot be printed directly.

### 4. Web3 Serialization with Blockchain and Non-Fungible Tokens

The traceability and serialization process discussed in the previous section contains many technological and governance issues that are summarized below:

- *Data retention*: all the actors of the supply chain have to keep and maintain the information about the product for many years.

On the one hand, this is certainly true for the producers that have responsibility over the electronic batch record, which contains the data of production, and on the other hand, the warehouse and the retailers also have the duty to append information about the status of the package (for instance, if products get damaged or compromised).

In any case, this turns into a responsibility to guarantee the final customer.

- *Data synchronization*: along with data retention, another important duty of the supply chain actors is to synchronize the status of the product with the central authority. This is a relevant activity that increases success in guarding against counterfeiting because the central authority can monitor the responsible actor of the package in any step of the distribution network.
- *Ownership*: this concept is tightly linked to the responsibility that each actor owns during the handling of the package in each step of the distribution network. In this sense, the responsible actor has to update the status of the product in order to guarantee the integrity of the distribution process.
- *Immutability*: the data characterizing any package (and its content) must not be manipulated and must describe only the information of traceability that has been appended to the product by the actor of each step of the distribution network.

With the technological infrastructure discussed in Section 3, the previous requirements are only partially tackled because in most of the current distribution networks, synchronization and verification with the central authority only occurs when the product enters in the distribution network. Potentially, before a product is dispensed to a patient (hospital, etc.) or to a consumer (pharmacy), a verification check should be performed so as to guarantee that the GTIN and the unique serial number of the item's packaging are valid and have been uploaded in the central authority's database by the real manufacturer of the product. Moreover, if the ability to synchronize the status of the packaging could occur at any step of the distribution network, the value of the traceability process could increase tremendously with incredible benefits for all stakeholders, including the patients. Unfortunately, this scenario is much more challenging because it would require a common software layer for each supply chain actor that, when integrated at the level of their information system (such as their ERP), offers the ability to communicate and synchronize with the central authority automatically.

Blockchain technology has key features that solve the previous issues [82] and, with the cutting-edge tools of Web3, there is a room to increase the adoption of a decentralized paradigm and, by this, improve the traceability process and simplify the integration among the actors of the distribution network. In order to interact with the blockchain (and related smart contracts), users have to operate by means of public and private keys that allow the basic operations of signing and retrieving information. The private key, also known as the secret key, is a variable in cryptography that is used with an algorithm to encrypt and decrypt data. It is personal and must not be given to anyone, otherwise all of the owner's assets in the blockchain can be stolen. Moreover, it allows the creation of multiple public keys that can be used as a public identity (i.e., public address) to operate with other entities. What Web3 offers is a common software layer for web-based applications (web apps) and

digital wallets that simplifies the interaction with the blockchain by also assigning private and public keys (and the related public addresses). This allows the reading and performing of complex business actions by means of smart contracts, which are pieces of software code stored and running in a blockchain. Additionally, smart contracts are identified by a public address once they are deployed in the blockchain; therefore, they can be publicly invoked by any other entity in the blockchain that has the rights to use it, according to the algorithm and rules coded in the smart contract.

Nowadays, web apps are becoming very popular because they can be used on smart devices connected to the Internet, such as smart mobiles or tablets, and such types of devices have the potentiality to replace proprietary scanners and readers because they are equipped with a camera, Bluetooth interface and, more recently, NFC readers. In regards to the traceability of drugs, in [58], a decentralized Web3 application for the Ethereum blockchain is presented. The authors discussed the main functions of the smart contract for handling the product lot and buying or selling a box, specifying that only the trusted entities are allowed to use the smart contract. This last concept is at the basis of non-fungible tokens (NFTs) [83] that, in the last two years, are ramping up to Web3 solutions that are popular among consumers and organizations. A non-fungible token represents a unique asset with an immutable identity in the blockchain that is used to create a unique user experience, encouraging interaction and stimulating interest in the brand and in the product [84]. Non-fungible tokens are different from cryptocurrencies; the latter can be traded or exchanged for one another while an NFT is unique and is characterized by its digital signature, which makes it impossible to be exchanged with another. With this point of view, the NFT can represent the perfect digital twin of any physical item distributed along a supply chain and, for the traceability process, this element provides a bulletproof solution against counterfeiting. This last concept will be explained and clarified in the next section that discusses a prototype application built for the traceability process of a generic pharmaceutical factory.

*4.1. Non-Fungible Token as Digital Twin of a Serialized Item*

As explained in Section 3, serialization requires that any parcel assembled by the packaging lines of a pharmaceutical factory has to be characterized by a unique identifier, in the form of a QR code that contains the relevant information for its identification and its traceability along the distribution network. Therefore, it can be possible to create a digital twin version of each QR code by minting an NFT in the blockchain.

As shown in Figure 5, an NFT is characterized by the following properties:

- *Uniqueness*: NFTs are cryptographic tokens providing a representation of unique assets with individual characteristics used to differentiate them from one another.
- *Authenticity*: NFTs provide a representation of real-world assets, establishing their authenticity. Authenticity is a key feature of an NFT because it ensures the uniqueness of NFTs.
- *Ownership*: NFTs are indivisible and can be only owned by the entity that has ownership of it.
- *Interoperability*: NFTs are stored in a smart contract in the blockchain. Due to the previous features, it becomes possible to use NFTs at different levels of granting access in Web3 applications.

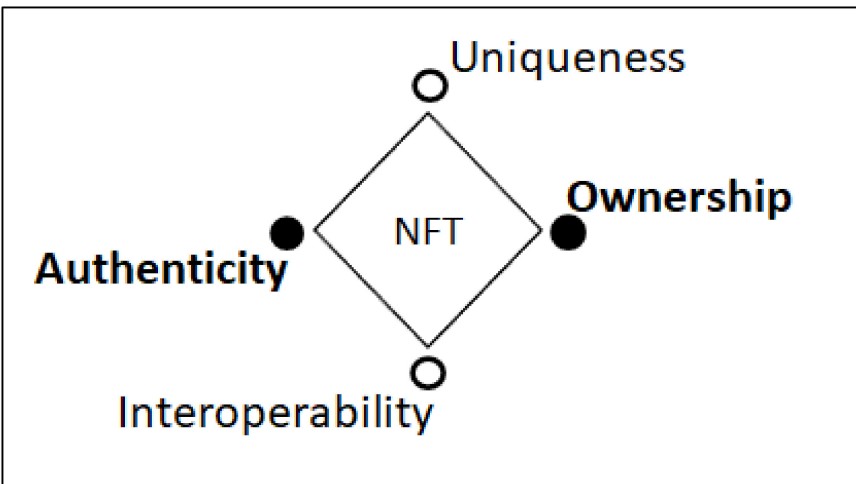

**Figure 5.** Main properties of a non-fungible token (NFT).

An NFT of a serialized item has to contain the following fields:

- *NFTId*: this field corresponds with the GTIN and serial number assigned by the serialization process. This unique ID is added as an index of the NFT in the smart contract. This field cannot be modified;
- *NFTSerialized* (GTIN, Serial Number, Lot Information, Expiration Date): this field contains the information of the serialized item, contained in the QR code of the label attached to the package. This field cannot be modified;
- *NFTProperties* (Commercial Name, Active Principle, Company Name, Description): this field cannot be modified;
- *Creator Public Address*: this field must be the public address of the pharmaceutical factory (corresponding to the GLN number of the company). This field cannot be modified;
- *Owner Public Address*: this corresponds with the public address of the actor of the distribution network that has ownership in that phase of distribution. This field can change during the process;
- *NFT Events* (<Id, OwnerAddress, Timestamp, Location, Event, Description>): this field represents the list of events that characterize the NFT lifecycle, where:
  - ○ Id: is an incremental integer (handled by the smart contract);
  - ○ Address: corresponds with the address of the owner of the NFT that performs the action characterizing the event;
  - ○ Timestamp: is the datetime of the event;
  - ○ Location (nullable): is the GPS coordinates where the event occurs;
  - ○ Event: is the type of event that has occurred with the NFT;
  - ○ Description (nullable): is additional information that can be attached to the event.

This field can grow in the number of items; once appended the item cannot be modified during the process.

- *ChildOf* (nullable): this field corresponds with the NFTId of the parent NFT used to handle the hierarchical packaging. This field can change (during the reconfiguration of the packaging) and is generally assigned after the minting;
- *Children* (NFTId[]):this field contains the list of children NFTIds that handle the hierarchical packaging (in case the NFT represents a second or third level of packaging). This field can change (during the reconfiguration of the packaging).

The smart contract that encrypts all the previous information must be codified in order to handle and strengthen the traceability process along the distribution network. In order to do that, as soon the packaging of a drug is complete and the QR code is applied

to the box, the NFT of the serialized item has to be minted, using the smart contract method MintNFT(NFTId, NFTSerialized, NFTProperties). For instance, let us assume that a pharmaceutical factory characterized by its GLN and a blockchain public address (e.g., 0xAABB . . . ..PP34) is producing and packaging a product with a GTIN (e.g., 3353..34885); the first event of the NFT is the "minting", with the following properties:

- NFTId: <next id of the smart contract>;
- SerializedItem: <3353..34885>, <serial number>, <lot number>, <date of expiration>;
- Creator Public Address: <0xAABB . . . ..PP34>;
- Owner Public Address: <0xAABB . . . ..PP34>;
- NFTEvent: [0, <0xAABB . . . ..PP34>, <hh:mm:ss dd/mm/yyyy>, <GPS Coordinate>, <minting>, <description of the drug packaged>].

In order to handle the hierarchical packaging, the smart contract has to contain a method for grouping the NFTs accordingly. Only the owner of the NFT can perform such types of operation, using the method of the smart contract GroupNFT(parentNFTId, childrenNFTId[]).

In general, the smart contract is coded in a way that only the owner can use the method for adding events to an NFT. In this way, at each step of the distribution network, the owner responsible for the serialized item has to update the information of the packaging (for instance, the pharmaceutical factory is the owner of a pallet as long as it is not dispatched to the next actor of the distribution network). The products can be, interchangeably, a pallet, a case box, or a single unit box of product as it depends on the type of shipping and actor of the distribution network (e.g., a warehouse is likely to handle pallets, a wholesaler, several case boxes, and a pharmacy, single unit boxes). Moreover, before the dispatching of the serialized item to the next actor of the distribution network, the owner has to use a specific method that appends the event of dispatch to the next owner in the NFTEvent list. This last method modifies the property Owner Public Address with the address of the next actor of the supply chain that, from that time on, will be in charge of handling the product.

Therefore, the smart contract has to reveal two more methods, which can only be used by the NFT owner:

- AppendEvent (Event, NFTId)
- UpdateOwner (Public Address, NFTId)

Moreover, it is important to highlight that, after the minting, the NFT is characterized by the Owner Public Address of the manufacturer. This makes it impossible for others to append information in the smart contract, although any actor can, in principle, retrieve the public information characterizing the NFT.

Finally, the method BurnNFT(NFTId) is used when the last owner administers (doctor) or sells (pharmacy) the product to the patient.

### 4.2. The Robust Traceability NFT Process

As explained in the previous section, when an item is serialized and minted, the only actor of the supply chain that can append information about the status of the item is the creator (who at that specific moment corresponds with the owner). This binding creates a powerful mechanism of anticounterfeiting, as explained in Figure 6, which shows the generic distribution process with the locking mechanism operated by the smart contract. In fact, the smart contract method UpdateOwner is a secure mechanism for granting the ownership token (T) among the actors of the distribution network. In the example in Figure 6, it is assumed that there are four actors in the distribution network. At the minting, the ownership of the NFT is the producer who is the pharmaceutical factory (Figure 6a).

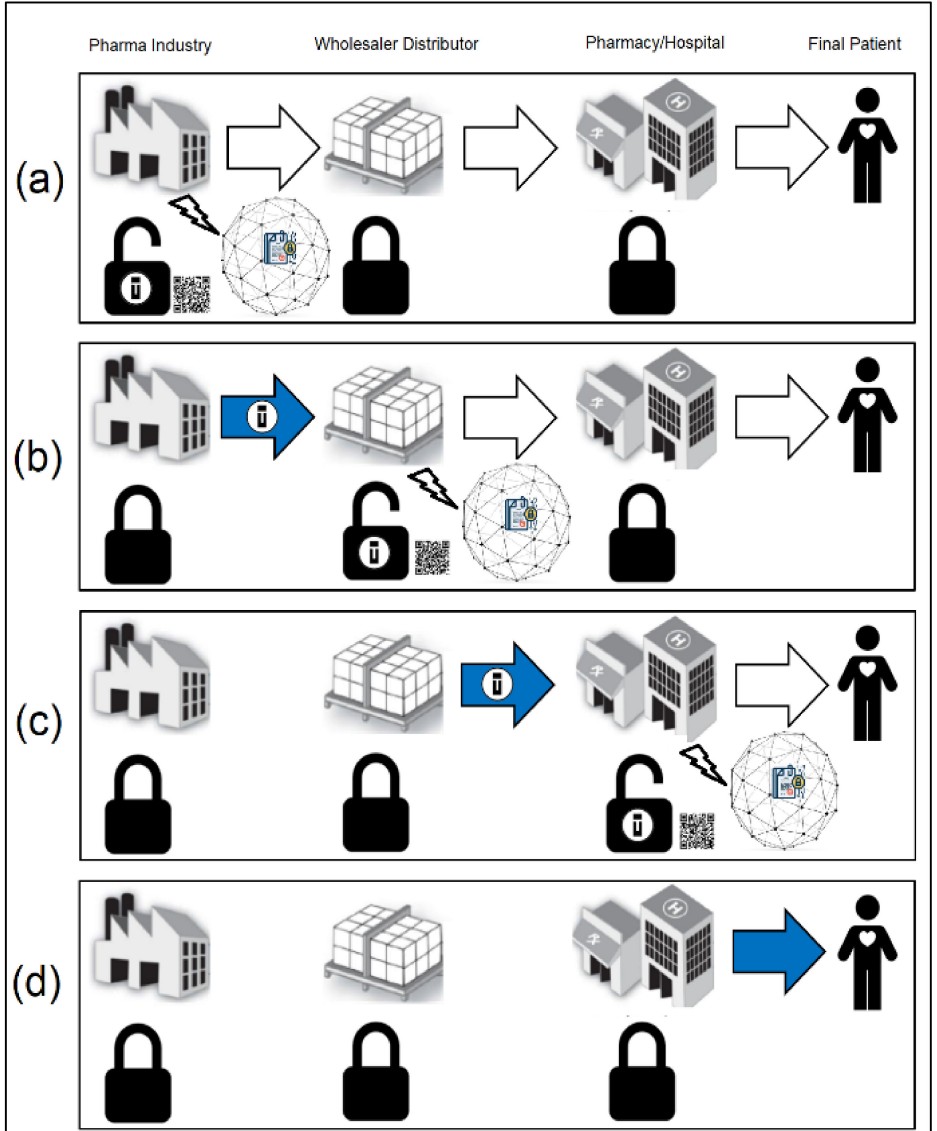

**Figure 6.** An example of robust track and trace process with NFT serialization: (**a**) the NFT is minted; (**b**) the ownership is passed the warehouse; (**c**) the ownership is passed to the hospital retailer; (**d**) the NFT is burnt as soon is administered to the patient.

As owner of the token, only the pharmaceutical factory can invoke the functions of the smart contract by means of its private key. In order to carry on with shipping the package to the next actor in the supply chain, the wholesaler distributor, the pharmaceutical factory has to invoke the function UpdateOwner in order to pass the ownership token to the wholesaler distributor, and modify the Owner Public Address of the NFT for the package.

Once the package is received by the wholesaler distributor (Figure 6b), a "delivered" event is appended to NFTEvent list in the smart contract, including, if needed, other information about the status of the package. Additionally, in this case, before the next shipment, the token owner has to pass the ownership to next actor, the hospital (or pharmacy) that, acting as the last actor of the distribution network, will update the information about the package (Figure 6c) with the event "ready for selling", which guarantees that the product can be administered (or sold) to the patient. Once the product is given to the patient, the NFT is burnt by transferring the ownership to an inaccessible address such that no one else will ever be able to modify the NFT information (Figure 6d).

The traceability process hereby described provides a powerful tool to the final client that verifies the history of the product and checks whether it is at the correct status to be

sold. In fact, during the entire production and distribution process, the transparency of the blockchain allows all actors to analyse the status of each item by scanning the QR code with a Web3 application. This allows the reduction and prevention of counterfeiting attempts, as shown in the example in Figure 7. For instance, if at a certain point of the distribution network a counterfeited product enters the distribution chain, the malicious path can be easily revealed. In fact, the next actor who is receiving the package will be able to check the status of the NFT and verify that the last event in the NFTEvent list does not show the delivered status in favour of the previous actor who owns the physical product, nor that this last actor was able to update the NFT in favour of the next owner. In the example in Figure 7, Patient A can verify that the product purchased was "ready to be sold", as certified by Pharmacy A, which is the legitimate selling point. On the other hand, Patient B can visualize the same information shown to Patient A, understanding that (i) the selling point is not Pharmacy B and (ii) that the product has already left the distribution network (because it was sold by Pharmacy A).

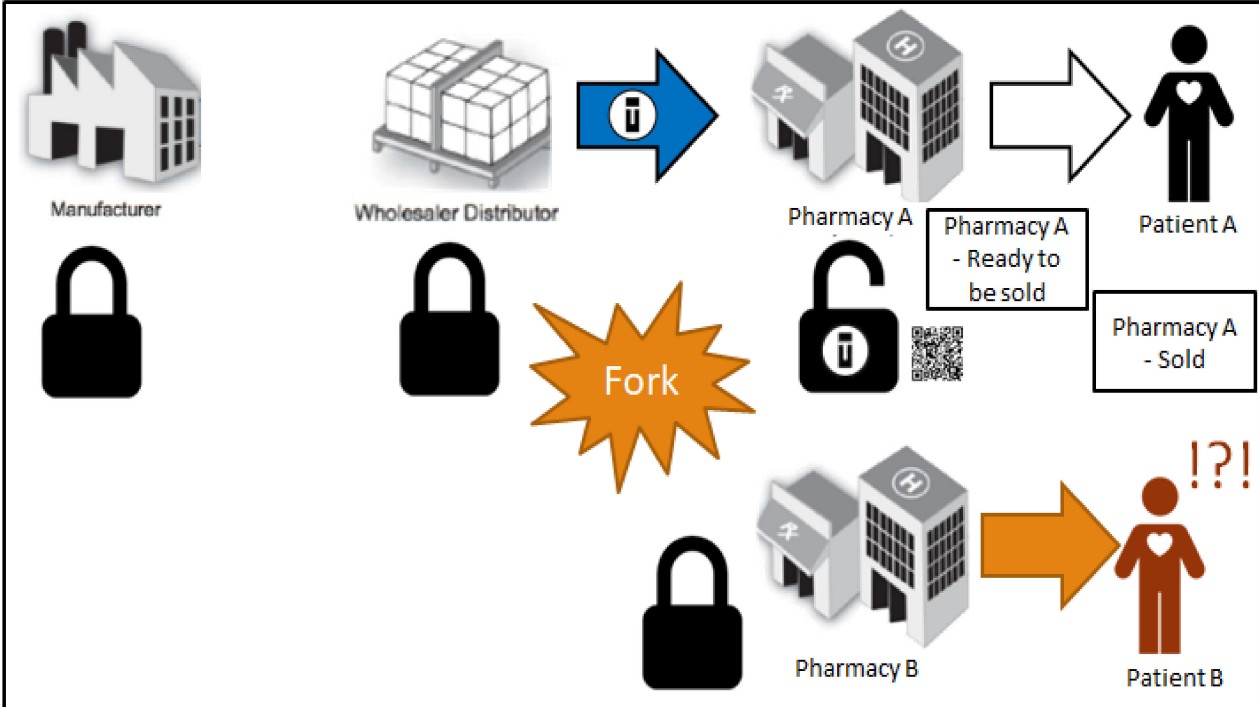

**Figure 7.** Use case of a counterfeit product that can be detected with NFT serialization.

The smart traceability process based on the proposed blockchain solution is shown in Figure 8. As it can be seen, it is similar to the one shown in [9] and the main change that has been made in this part of the process arises from the use of a publicly permissioned blockchain, as it will be seen in the next subsection. Any time a package transits into a new phase, the operations of (1) tracing info, (2) updating status, and (3) enabling the next actor of the distribution network must be undertaken.

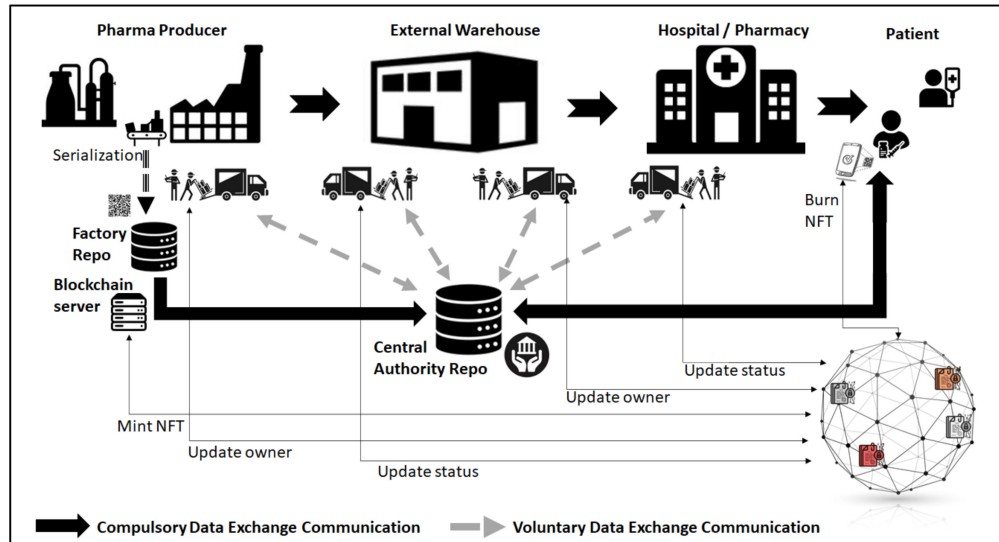

**Figure 8.** Track and trace process with a central authority and blockchain serialization.

This guarantees the tracking, the synchronization, and the integrity of all the activities in the blockchain. Once deviations between the "real world" and the ledger on the blockchain occur, counterfeit actors can be identified. In any case, this process does not replace the one dictated by the 2011/62/EU Directive [10], but it can be executed in parallel, exploiting the DAPP prototype developed, which is the focus of the next subsection.

Figure 9 shows the architecture of the blockchain serialization system deployed from the manufacturer enterprise. It can be observed that at Level 4, a blockchain server is added to communicate with the serialization manager. As soon a unit box is labelled with a unique identifier (as required by the serialization specification) and stored in the enterprise database, it is possible to gather the relevant information from the serialization manager in order to mint the NFT of the packaging. As explained, this task is performed by the blockchain server that, as it is shown, communicates directly with the blockchain network.

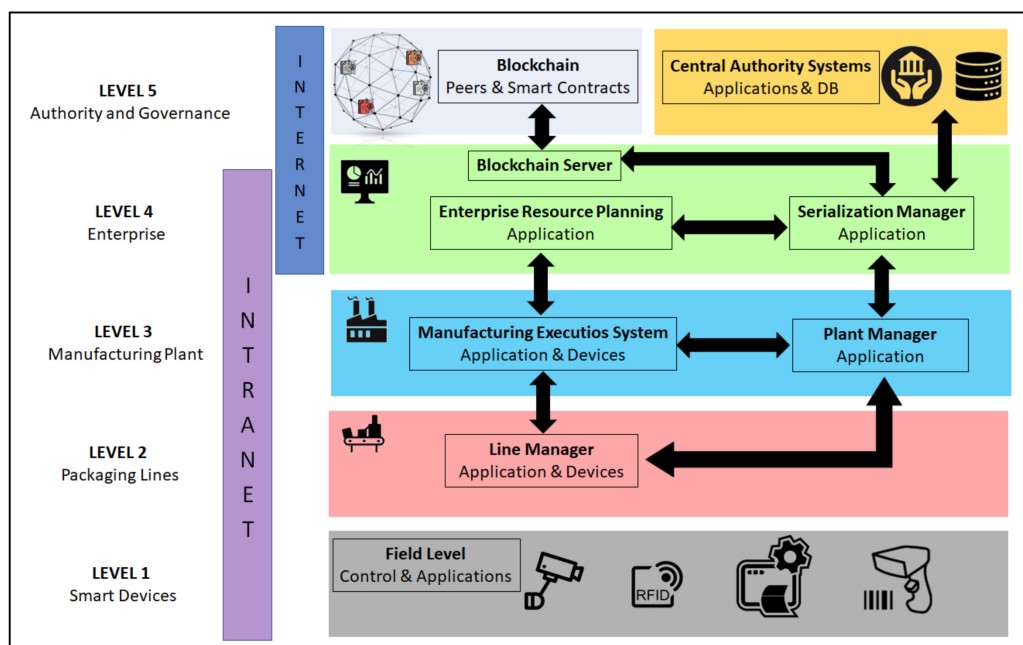

**Figure 9.** Architecture of the information system for the blockchain serialization with NFT.

The data flow process of the blockchain serialization for a unit box (primary packaging) and a case Box (secondary packaging) is described in Figure 10. Once the NFT is minted and stored in the blockchain, it can be read and eventually updated using the unique identifier QR code assigned by the smart contract. In fact, any time the owner needs to update some information about the status of the product, an event is created and appended in the structure NFTEvent list of the NFT. In this way, the history of the NFT associated with the QR code of a packaging unit can be tracked by scanning it with a progressive decentralized web application.

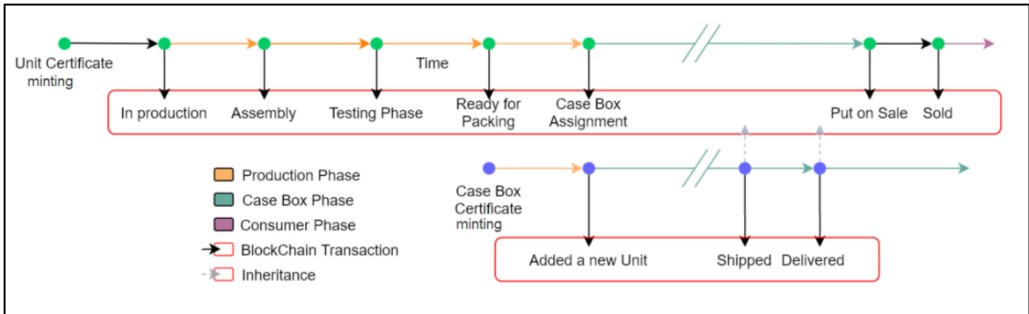

**Figure 10.** Process flow of pharmaceutical factory packaging using the blockchain serialization.

### 4.3. Architecture and Implementation of the NFT Track and Trace Solution

The smart contract of the proposed NFT track and trace prototype solution, has been deployed in a test environment of the VeChain Thor blockchain, a publicly permissioned proof of authority (PoA) consensus blockchain that guarantees higher scalability and performance than traditional proof of work (PoW) protocols [85]. This blockchain is provided by the VeChain Foundation, which lists 101 authorized master nodes (known also as Thrudheim) that are in charge of providing the validation of the blockchain transaction. The ownership of the master nodes belongs to several private and public companies, organizations, and individuals. To guarantee the stability of the ecosystem, the blockchain ecosystem can count on what are called economic nodes. The main reason for the adoption of this blockchain is that the VeChain ecosystem aims at building a trust-free and distributed business ecosystem platform to enable transparent information flow, efficient collaboration, and high-speed value transfers [86]. These key elements have made VeChain one of the most valuable blockchains to use for supply chain and track and trace. Moreover, the foundation provides a user-friendly Web3 connector, Sync2, that can be installed as a plugin in most well-known browsers.

Figure 11 shows the progressive web decentralized application developed for this prototype. For each phase of the lifecycle of a product, this application shows the status, the timestamp of the transaction, the public address of the owner who entered the data (who has signed the transaction), and a link ("View Transaction") that opens the public blockchain explorer containing all the details of the corresponding transaction. Two main actions can be performed as the owner, for each NFT: the creation of a new event and the handover to a new owner.

Figure 12 presents the infrastructure of the NFT track and trace solution, showing the main components that constitute it. As discussed previously, the generic pharmaceutical manufacturer runs the blockchain server (integrated with the serialization equipment) to mint the NFTs, interacting directly with the NFT smart contract of the blockchain, by means of its public address. The generic user's user device, by means of a mobile device connected to the Internet, can scan the QR code of the NFT, redirecting the user to the progressive web decentralized application that is hosted in the DAPP repository. The DAPP is coded to unlock or lock the granted access of the user who—due to the Sync2 plugin—interacts with the smart contract.

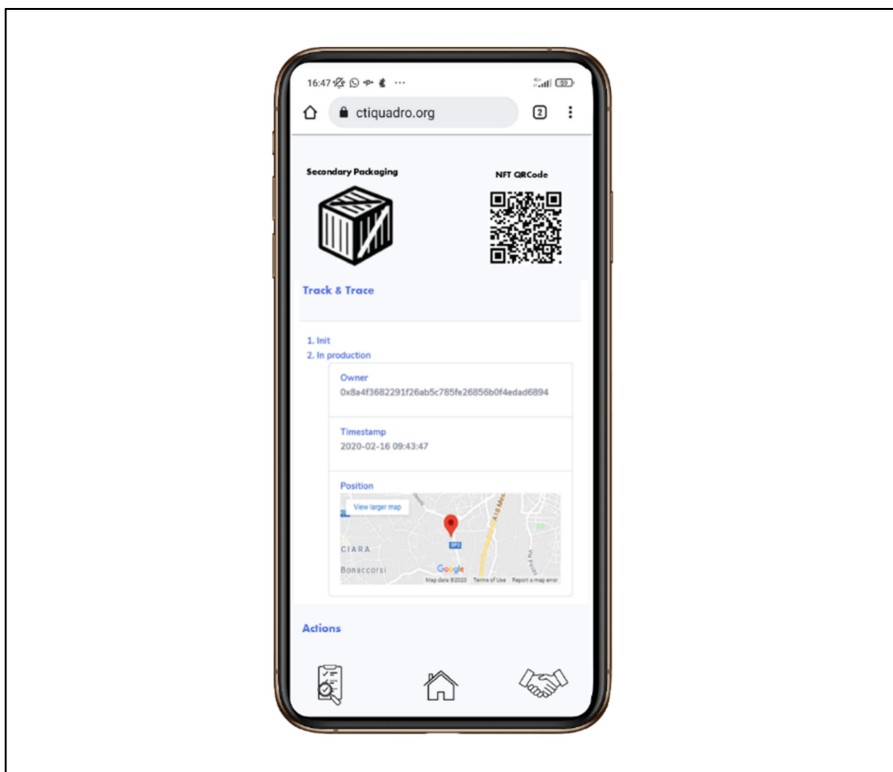

**Figure 11.** Prototype of the DAPP running within the VeChain Thor blockchain test network.

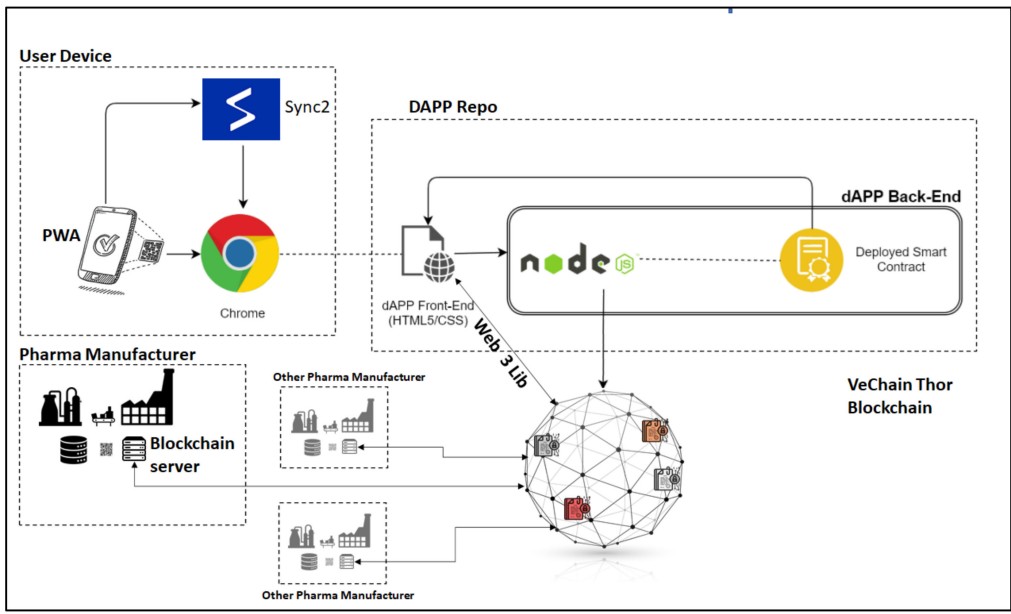

**Figure 12.** High-level SW/HW Infrastructure of the NFT Track & Trace ecosystem.

For the NFT track and trace, in order to perform as an interoperable solution that complements the serialization process, three major requirements must be fulfilled:

(1) Each actor of the supply chain (who is going to become an active owner of a package) has to be identified with its VeChain public address (basically, this is not necessary for the final consumers);

(2) Each actor of the supply chain has to keep VET or VeThor tokens in a wallet as they are required to sign the transactions and update the status of the package;

(3) Each actor of the supply chain has to know the public addresses of the companies involved in the distribution network in order to pass the NFT to the correct owner. This last requirement can be easily maintained by using the track and trace decentralized web application, provided that the companies register themselves autonomously (at the moment they retrieve their public address or even later) using the Sync2 plugin.

## 5. Conclusions

In these last few years, the industrial makeup has seen a tremendous boost of innovation. Under the hat of Industry 4.0 (I4.0) initiatives, technological and digital solutions have been integrated at different levels of a company, transforming enterprises into smart organizations able to improve the effectiveness of their management, the sustainability of their production, and their distribution processes, and to increase communication with all the stakeholders of the supply chain.

In this paper, a thorough review of I4.0 applications has been carried out, describing advantages and barriers for different industry sectors. Among those, a very interesting I4.0 business use case, the serialization of the pharmaceutical drugs, has been summarized highlighting its main elements. Due the latest good manufacturing practice (GxP) directives, in the previous years, its implementation has become crucial for this industry sector, introducing a stricter regulation framework for the distribution and commercialization of pharmaceutical products worldwide and forcing drug manufacturers to renew their production lines with smart Industry 4.0 equipment. Then, it was pointed out that serialization is at the basis of the track and trace process, along the entire pharmaceutical distribution network up to the final consumer.

The main issues of the centralized governance architecture were discussed, providing the motivations for a decentralized solution able to compliment the standard track and trace process and overcome its inherent limitations. Due to the I4.0 intercommunication solutions, an innovative solution based on non-fungible tokens (NFTs) has been proposed; specifically, the solution discussed in this paper builds upon the generic serialization technology enforced by the pharmaceutical manufacturers and can be easily plugged in without interfering or modifying the standard serialization process. Moreover, it was shown how such an approach can strengthen the track and trace process, improve the communication among the supply chain stakeholders, and increase trust with the final consumers.

The proposed solution was conceived to run on a test environment of a publicly permissioned blockchain, the VeChain Thor blockchain, and all the basic SW/HW components required to test a prototype application were discussed: the decentralized progressive web app, the blockchain server that synchs on the pharmaceutical manufacturer's serialization manager, and the NFT smart contract.

Objectives of the next research activities are to complete and implement the NFT track and trace prototype in mainnet, start a collaboration with local enterprises, and test the feasibility of the solution with real stakeholders in order to understand what else can be done to improve this process, track some relevant metrics to analyse bottlenecks, and provide valid remediations.

**Author Contributions:** Conceptualization, F.C. and C.S.; methodology, A.S. and L.M.O.; software, F.C. and A.S.; validation, D.D., D.G. and C.S.; formal analysis, F.C.; investigation, F.C.; resources, D.G.; data curation, D.D.; writing—original draft preparation, F.C.; writing—review and editing, C.S.; visualization, L.M.O.; supervision, D.G.; project administration, C.S.; funding acquisition, D.G. All authors have read and agreed to the published version of the manuscript.

**Funding:** This research has been partially supported by project Biotrak Azione 1.1.5-PO FESR 2014–2020 n. prog. 08SR1091000150-CUP G69J18001000007.

**Informed Consent Statement:** Not applicable.

**Data Availability Statement:** Data sharing not applicable.

**Conflicts of Interest:** The authors declare no conflict of interest.

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
