# Peer review of "A Non-Fungible Token Solution for the Track and Trace of Pharmaceutical Supply Chain"

_applsci, doi:10.3390/app12084019_

Round 1

Reviewer 1 Report

This is a paper describing the technical characteristics of a non-fungible token architecture in a prototypical stage to be employed in pharmaceutical industry. Although the authors did not carry out an analysis, for example, of the use of the proposed tool, as it is in a prototypical stage, the way they followed to obtain the prototype is very well detailed and illustrated. 

I believe that it is a valid communication so that the authors can obtain feedback from the community interested in the topic, to continue the necessary developments for the consolidation of the proposed system. The study also synthesizes interesting knowledge for the interested community, being able to act as an initial guide for the concepts that were applied.

The literature review presented in section 2 of the work is quite comprehensive, covering topics such as IoT, CPS, M2M, Cloud Computing, Big Data, Smart Factory, ERP, Augmented Reality, etc. All these topics consistent with the Industry 4.0 perspective. The Authors created a great synthesis of these topics, to help clarify to readers about the insertion context of the reported research.

Below I make some suggestions to help authors improve their text:

a)    In the introduction the authors define: "This paper describes a practical use case with a prototype application that helps to understand how to use it and its benefits."

As I understand from reading the material, the real objective of the paper can be stated as "to present the concepts and architectural elements necessary to support the non-fungible token solution, culminating in the presentation of a use case with a prototypical application". 

b)    In section 2.2, line 124, it may be better to separate two sentences with a period. It currently looks like this:

Cyber Physical Systems (CPSs) refers to the integration of computing and physical processes, can be divided in two elements (i) a network of objects and systems [...].

I believe that, to make the text better cadenced, it is better like this:

Cyber Physical Systems (CPSs) refers to the integration of computing and physical processes. These systems can be divided into two elements: (i) a network of objects and systems [...].

c)    I ask if there are no works related to the study being reported in this manuscript. It would be interesting to carry out a comparative analysis, at a conceptual level, between this study and others that contain similar proposals. In Table 1 (page 3) the authors presented a synthesis of the concepts presented in their literature review, with strengths, disadvantages, sectors, and works, but I did not see a comparison with papers presenting proposals like theirs.

d)    Still about Table 1, I think the caption “This is a table. Tables should be placed in the main text near to the first time they are cited” is the original in the journal’s template. the authors forgot to update this as per what this table represents.

e)    Section 4 of the work is very rich in details, throughout the discussion built by the authors, including impacts on the perspective of customers who will be able to make use of the system, when finalized.

Throughout this discussion, it is important that authors use related literature to support the ideas presented. In fact, I noticed that from page 8 onwards, the authors only carry out 4 more cited works (81, 82, 83 and 84). In a text whose objective is to present concepts related to a system architecture, it is essential that the theoretical discussion of the components be extended with more citations, to reinforce the ideas presented. Perhaps the comment I made in c) can help in this regard.

Author Response

  • Reviewer’s Comment: This is a paper describing the technical characteristics of a non-fungible token architecture in a prototypical stage to be employed in pharmaceutical industry. Although the authors did not carry out an analysis, for example, of the use of the proposed tool, as it is in a prototypical stage, the way they followed to obtain the prototype is very well detailed and illustrated. I believe that it is a valid communication so that the authors can obtain feedback from the community interested in the topic, to continue the necessary developments for the consolidation of the proposed system. The study also synthesizes interesting knowledge for the interested community, being able to act as an initial guide for the concepts that were applied. The literature review presented in section 2 of the work is quite comprehensive, covering topics such as IoT, CPS, M2M, Cloud Computing, Big Data, Smart Factory, ERP, Augmented Reality, etc. All these topics consistent with the Industry 4.0 perspective. The Authors created a great synthesis of these topics, to help clarify to readers about the insertion context of the reported research.

Authors’ Reply: Authors are very grateful for this comment.

Below I make some suggestions to help authors improve their text:

  • Reviewer’s Comment: In the introduction the authors define: "This paper describes a practical use case with a prototype application that helps to understand how to use it and its benefits." As I understand from reading the material, the real objective of the paper can be stated as "to present the concepts and architectural elements necessary to support the non-fungible token solution, culminating in the presentation of a use case with a prototypical application". 

Authors’ Reply: Authors thank for this kind suggestion. The paragraph has been updated accordingly.

  • Reviewer’s Comment: In section 2.2, line 124, it may be better to separate two sentences with a period. It currently looks like this:

Cyber Physical Systems (CPSs) refers to the integration of computing and physical processes, can be divided in two elements (i) a network of objects and systems [...].

I believe that, to make the text better cadenced, it is better like this:

Cyber Physical Systems (CPSs) refers to the integration of computing and physical processes. These systems can be divided into two elements: (i) a network of objects and systems [...].

Authors’ Reply: Authors thank for this kind suggestion. The paragraph has been updated accordingly.

  • Reviewer Comment: I ask if there are no works related to the study being reported in this manuscript. It would be interesting to carry out a comparative analysis, at a conceptual level, between this study and others that contain similar proposals. In Table 1 (page 3) the authors presented a synthesis of the concepts presented in their literature review, with strengths, disadvantages, sectors, and works, but I did not see a comparison with papers presenting proposals like theirs.

Authors’ Reply: In the paper [57] which was cited as one of the main benchmark for our paper, the main novelty that we are proposing is the use of public permissioned blockchain which is more appropriate than the Ethereum Blockchain (a public permissionless blockchain):

“In [57] a decentralized solution for the track and trace of the pharmaceutical drugs along the supply chain is presented. This solution is based on the Ethereum blockchain which is the most famous public permissionless blockchain. Due to this great feature, the Ethereum blockchain is used  and abused (as it is at the basis of tons of applications), and this decreases its performance while increasing the transactions costs. For this reason, the Ethereum blockchain is not the  most appropriate for a supply chain ap-plication and stakeholders should look for a different blockchain platform. As it will be shown in the next sections, in this paper the VeChain Thor Blockchain, a public per-missioned blockchain, will be used.”

Moreover, literature does not present, at the state of the art any reference to the use of Non-Fungible Token in combination with the blockchain for the supply chain.

  • Reviewer Comment: Still about Table 1, I think the caption “This is a table. Tables should be placed in the main text near to the first time they are cited” is the original in the journal’s template. the authors forgot to update this as per what this table represents.

Authors’ Reply: authors thank you the reviewer for this comment. We have made the correction in order to attribute a correct label to the Table 1.

  • Reviewer Comment: Section 4 of the work is very rich in details, throughout the discussion built by the authors, including impacts on the perspective of customers who will be able to make use of the system, when finalized. Throughout this discussion, it is important that authors use related literature to support the ideas presented. In fact, I noticed that from page 8 onwards, the authors only carry out 4 more cited works (81, 82, 83 and 84). In a text whose objective is to present concepts related to a system architecture, it is essential that the theoretical discussion of the components be extended with more citations, to reinforce the ideas presented. Perhaps the comment I made in c) can help in this regard.

Authors’ Reply: authors thank you the reviewer for this comment. We have added some relevant reference to strengthen some important points and edited the References that contained, by mistake some duplicate. Moreover, several references that were previously cited in the paper have been reused to reinforce some points in Section 4, as you suggested to do. See below the major changes:

  1. “The Blockchain technology has the key features to solve the previous issues [82]”

[82] Sylim, P.; Liu, F.; Marcelo, A.; Fontelo, P. Blockchain technology for detecting falsified and substandard drugs in distribution: Pharmaceutical supply chain intervention. JMIR Res. Protoc. 2018, 7, doi:10.2196/10163.

  1. As regard to the traceability of drugs, in [58] a Decentralized Web3 Application for the Ethereum Blockchain is presented. Authors discuss the main functions of the smart contract for handling the product lot and buy/sell a box, specifying that only the trusted entities are allowed to use the smart contract. This last concept is at the basis of Non-Fungible Tokens (NFTs) [83] that, in these last two years, are giving a ramp-up to Web3 solutions giving popularity among consumers and organizations.”

[58] Musamih, A.; Salah, K.; Jayaraman, R.; Arshad, J.; Debe, M.; Al-Hammadi, Y.; Ellahham, S. A blockchain-based approach for drug traceability in healthcare supply chain. IEEE Access 2021, 9, doi:10.1109/ACCESS.2021.3049920.

[83] Wang, Q.; Li, R.; Wang, Q.; Chen, S. Non-Fungible Token (NFT): Overview, Evaluation, Opportunities and Challenges. arXiv Prepr. arXiv2105.07447 2021.

  1. “The smart traceability process based on the proposed blockchain solution is shown in Figure 8. As it can be seen it is similar to the one shown in [9], and the main change that has been made in this part of the process arises from the use of a public permissioned blockchain, as it will be seen in the next subsection of this chapter.”

[9] Chiacchio, F.; Compagno, L.; D’Urso, D.; Velardita, L.; Sandner, P. A decentralized application for the traceability process in the pharma industry. Procedia Manuf. 2020, 42, 362–369, doi:10.1016/j.promfg.2020.02.063.

  1. d) “This guarantees the tracking, the synchronization and the integrity of all the activities in the blockchain. Once deviations between the “real world” and the ledger on the blockchain occur, bad actors can be identified. In any case, this process does not replace the one dictated by the 2011/62/EU [10], but it can be executed in parallel exploiting the DAPP prototype developed, object of the next subsection.”

[10] Bhaskaran J; Venkatesh M P Good Storage and Distribution practices for Pharmaceuticals in European Union. J. Pharm. Sci. Reserach 2019, 11, 2992–2997.

Reviewer 2 Report

The manuscript presents interesting data on industry 4.0 applications in the pharmaceutical industry, and a detailed presentation of solutions based on Non-Fungible Tokens (NFTs), which can improve the Track & Trace capability of the standard serialization process is presented. The Reviewer read the manuscript thoroughly and considered it suitable for publication as insight into the neglected issue when presenting the practical usage case with a prototype application that helps to understand how to use it and its benefits.

Author Response

Authors thank you the reviewer for this comment and his/her kind review.

Reviewer 3 Report

I propose to the authors of this paper to expand their research to both vertical and horizontal integration, networking within the company from the process through the operational level to the level of the production process, IT system connectivity at all levels.

Author Response

Authors thank you the suggestion. As explained in the conclusions, what the reviewer is proposing is already in our plan and will be carried out in the next research step. In fact, the integration within a company will be part of the next activities also in order to understand the quickness of a such integration and the benefit also from the operational point of view:

“Objective of the next research activities is to complete and implement the NFT Track & Trace prototype in mainnet, start a collaboration with local enterprises and test the feasibility of the solution with real stakeholders, in order to understand what else can be done to improve this process, track some relevant metrics to analyse bottlenecks and provide valid remediations.”

Round 2

Reviewer 1 Report

At this point, I believe the manuscript is ready for acceptance.
From my point of view, the main thing to be adjusted was the items on related studies and bibliographic support in section 4.

Regarding the related studies, the authors provided the necessary clarifications, pointing out that the literature does not present any state-of-the-art references on the use of the proposed combination in their work.

On the bibliographic complement of Section 4, the authors indicated the new citations inserted.